# Application of MaxEnt Modeling and HRM Analysis to Support the Conservation and Domestication of *Gevuina avellana* Mol. in Central Chile

**DOI:** 10.3390/plants11202803

**Published:** 2022-10-21

**Authors:** Mario René Moya-Moraga, César Pérez-Ruíz

**Affiliations:** 1Doctoral Program in Biotechnology and Genetic Resources of Plants and Associated Microorganisms (02E4), Polytechnic University of Madrid (UPM), University City, 28040 Madrid, Spain; 2Department of Biotechnology, Faculty of Natural Sciences, Mathematics and the Environment (FCNMM), Metropolitan Technological University (UTEM), Ñuñoa 7750000, Chile; 3Department of Biotechnology and Plant Biology, School of Agricultural, Food and Biosystems Engineering, Polytechnic University of Madrid (UPM), University City, 28040 Madrid, Spain

**Keywords:** *Gevuina avellana* Mol., high-resolution melting analysis (HRM), EST-SSR markers, maximum entropy modeling (MaxEnt)

## Abstract

The Chilean hazelnut (*Gevuina avellana* Mol., *Proteaceae*) is a native tree of Chile and Argentina of edible fruit-type nut. We applied two approaches to contribute to the development of strategies for mitigation of the effects of climate change and anthropic activities in *G. avellana*. It corresponds to the first report where both tools are integrated, the MaxEnt model to predict the current and future potential distribution coupled with High-Resolution Melting Analysis (HRM) to assess its genetic diversity and understand how the species would respond to these changes. Two global climate models: CNRM-CM6-1 and MIROC-ES2L for four Shared Socioeconomic Pathways: 126, 245, 370, and 585 (2021–2040; 2061–2080) were evaluated. The annual mean temperature (43.7%) and water steam (23.4%) were the key factors for the distribution current of *G. avellana* (AUC = 0.953). The future prediction model shows to the year 2040 those habitat range decreases at 50% (AUC = 0.918). The genetic structure was investigated in seven natural populations using eight EST-SSR markers, showing a percentage of polymorphic loci between 18.69 and 55.14% and low genetic differentiation between populations (Fst = 0.052; *p* < 0.001). According to the discriminant analysis of principal components (DAPC) we identified 10 genetic populations. We conclude that high-priority areas for protection correspond to Los Avellanos and Punta de Águila populations due to their greater genetic diversity and allelic richness.

## 1. Introduction

*Gevuina avellana* Mol. is a native tree of Chile and Argentina, monotypic of the genus *Gevuina* (*Proteaceae*) [1], also known as Chilean hazelnut, Chilean nut, Gevuina nut, and Chilean wildnut [2]. Its habitat is restricted to the subantarctic forests of southern South America, between 35° and 45° S, from sea level to 1500 m altitude. In Central Chile (Maule Region) it is currently confined to forest fragments as a companion species to dominant species in the different Andean sectors (the Andes and Coastal mountains) [3]. *G. avellana* is an evergreen, semi-tolerant species that is distributed in areas of Mediterranean, temperate humid or rainy climate, where precipitation varies between 500 and 4000 mm per year, and temperatures range from 30 °C in summer to −8 °C in winter [3,4] and adapts to various conditions of competition, light, and soil, prefers highly organic and well-drained volcanic soils, although it will also grow well on almost swampy and shallow, eroded, acidic soils [5]. The reproductive cycle lasts two years and comprises a differentiation stage in which the buds are defined irreversibly to vegetative or reproductive buds that occurs in autumn of year 1 and a flowering stage that occurs from summer of year 1 until the end of April of year 2 [6]. The pollination is entomophilous and self-incompatible, and the species can reproduce by seeds or through various methods of vegetative propagation [3,6]. Its fruit is an edible walnut that gets blackish when it ripens (from February to April). Its leaves are compound, perennial, alternate and imparipinnate. *G. avellana* inflorescence appears as a long white or pinkish flower cluster with approximately 50 hermaphrodite flowers, of which 5 to 7 get transformed into fruits. *G. avellana* have proteiform roots, and its trunk can grow straight or in branched and can reach heights close to 20 meters or more (Figure 1) [7]. The Chilean hazelnut has been traditionally consumed since the pre-Hispanic period, its original name in the Mapuche language is Gevuín or Gneufén (ñeufen), which means “eye that sees many flowers” concerning its abundant and beautiful flowering [8]. Spanish colonizers named it avellano, due to similarity of its fruit to the European hazelnut (*Corylus avellana* L.), but both species have no phylogenetic relationship, *G. avellana* belongs to the *Proteaceae* family, whereas *C. avellana* corresponds to the *Betulaceae* family [6]. In Chile, its cultivation is considered rare; the first antecedents of experimental plantations of *G. avellana* in Chile date from approximately 1970 onwards [9]. Some of these are maintained to this day, as is the case of an experimental orchard corresponding to the Genetic Improvement Program for *Gevuina avellana* Mol. located in Southern Chile (Valdivia, 39°45′ S–73°15′ W), where clones of edible walnuts of high productivity and quality have been developed, clones propagated mainly by cuttings [10,11,12,13,14,15,16]. Regarding its exploitation and production, it is based solely on collecting wild fruits from natural forest formations, and its fruits are marketed in local, regional, and national markets [17]. At the fruit harvest, 70% of the collectors belong to indigenous races; in general, it is farmers who have a subsistence economy and that represents the highest poverty rates in the country [18]. A phylogenetic study allocates the genus *Gevuina* together with seven other genera in the *Gevuinina* subtribe within the *Macadamieae* tribe [19]. The *Macadamia* genus also belongs to the *Proteaceae* family, and it corresponds to one of the few international food crops domesticated from the Australian flora and cultivated in tropical countries due to the quality of its fruits in terms of the high amount of oils they contain [20]. The Chilean hazelnut fruits are equivalent to the Macadamia nuts, having similar oils and nutritional contents [21,22,23]. *G. avellana* has been considered as an alternative to the macadamia crop in cold areas, with the possibility of getting a position in the international market with the same success [4]. *G. avellana* oil contains tocopherols (alpha and beta), especially alpha or vitamin E, that play a protective role in the cell membranes of the brain, anti-cholesterol, and retarder of Alzheimer’s symptoms [24]. Various products and by-products can be obtained from the fruit (e.g., coffee, cookies for celiacs, delicacy and honey cream) [25,26,27,28,29], both for human consumption, as well as for use in the cosmetic and medicinal industry [30,31,32,33]. The abundant inflorescence of the Chilean hazelnut is very attractive to bees, allowing the production of hazelnut honey [34,35].

Native plants are one of our most powerful tools for protecting and conserving healthy and resilient ecosystems. Successful native plant restoration facilitates the conservation of innumerable species interactions, such as between plants, their consumers, and their pollinators, that provide the foundation of ecosystem services upon which humans and wildlife depend [36]. When the Global Strategy for Plant Conservation (GSPC) was launched in 2002 it was recognized that plants are essential for the functioning of the planet and vitally important to support human livelihoods [37]. Native plants are often promoted to answer various environmental sustainability issues [38]. The United Nations Assembly established 17 Sustainable Development Goals (SDGs) for the year 2030 and has included goals related to restoration and reforestation on its agenda [39]. A successful reforestation program should consider the environmental and genetic factors that influence plant performance [40,41,42]. Climate is one of the critical factors influencing vegetation’s type and distribution globally [43]. It is therefore essential understanding the far- reaching effects of climate change and the associated anthropogenic impacts [44].

The maximum entropy modeling (MaxEnt) is a multivariate approach that estimates the distribution of a species by finding the probability distribution of maximum entropy, subject to constraints representing our incomplete information about the distribution [45]. The maximum entropy method (MaxEnt) is an effective algorithm for modeling the ecological and spatial distribution of species. MaxEnt models are usually based on WorldClim climate data obtained by interpolating the average monthly climate data of planet weather stations [46]. The niche model can be used to assess and predict the effect of climate change on plants [43]. Ecological niche modeling has become the main tool for modeling species distributions [47]. MaxEnt and the Geographical Information System (GIS) have become powerful and versatile tool used in different areas of knowledge, which delivers fast, reliable, and accurate results, whose application is widely reported, among which are: Territorial and impact studies on environmental protection, precision agriculture, ecological restoration and/or conservation studies, preparation of species distribution maps, evaluation of spectral signatures of photosynthetically active vegetation [48,49,50,51,52]. To determine the performance of the model, the “Area under the ROC Curve” (AUC) value is obtained by means of receiver operating characteristic (ROC) analysis. AUC has a direct relationship with model performance, and it is an excellent index to evaluate model performance. The value of the area under the Receiving Operator Curve (AUC) ranges from 0 to 1 [51]. The Jackknife testing represents the permutation-based importance of explanatory variables. MaxEnt Model enables the evaluation of distribution potential current and Future it is species [49]. The importance of the Shared Socio-Economic Pathways (SSPs) is to include new scenarios in climate change research. The Socio-economic scenarios used to derive emissions scenarios without (baseline scenarios) and with climate policies (mitigation scenarios). SSPs aim not directly at decision-makers but at climate change analysts preparing climate policy analysis based on the SSPs [53]. The SSP1 (Sustainability) scenario is oriented toward the development of environmentally friendly technologies; in the SSP2 (Middle) scenario, the social, economic, and technological trends of the historical patterns are maintained; in SSP3 (Regional rivalry) in this scenario, countries focus on achieving energy and food security goals within their own regions, decrease investments in education and technological development and SSP5 (Fossil-fueled development) this scenario is driven by the economic success of industrialized economies that combine the exploitation of abundant fossil fuel resources [54].

Monitoring global biodiversity from space through remote sensing geospatial patterns has a high potential to add to our knowledge acquired by field observation. Nevertheless, some essential biodiversity variables (EBVs) are not directly measurable by remote sensing from space, specifically the EBV class genetic composition [55]. To assess genetic diversity, sufficiently variable genetic markers are required so that can be applied to a large number of samples efficiently and profitably [56]. Due to their high reproducibility, multiallelic nature, and codominant inheritance, SSR molecular markers are widely used in plant germplasm identification, genetic diversity, genetic linkage map construction, gene mapping and cloning, and in quantitative trait loci analysis (QTLs) [57]. Molecular markers are used in many plant species due to their stable and highly polymorphic nature [58]. This approach has been applied widely to analyze genetic diversity with molecular markers, including SSRs, in several plant species [59,60,61,62,63]. The design of SSR molecular markers can be developed using information contained in genomic DNA (gSSRs) or expressed sequence tags sequences (ESTs). EST–SSR markers have many advantages compared to gSSRs, including their locus-specificity to the expressed genes and potential to be used as functional markers for association studies of candidate genes with phenotypic variations and comparative studies for genetically related different species [64] and because they are derived from transcripts, they are useful for assaying the functional diversity in natural populations or germplasm collections [65]. EST–SSR molecular markers have been developed for many plant species [66,67,68,69]. Over the last years, High-Resolution Melting (HRM) analysis has acquired great importance due to its simplicity, flexibility, sensitivity, and specificity [64,70]. HRM means of screening sequence variation post Polymerase Chain Reaction (PCR), provides rapid insights into the levels of sequence variation among samples through melt curve clustering [56]. The HRM method measures the fluorescence reduction of intercalating dye in the process of dissociation of double-stranded DNA, consisting of amplifying small DNA fragments (15–300 pb), denaturing and re-naturalizing them in a controlled way, using fluorophores and specialized real-time equipment that allows following the kinetics of renaturation of the strands. This kinetics is more or less specific and serves to identify the different allelic variants [71]. Despite its apparent usefulness, the use of HRM in conservation genetics study in native plants is practically unknown [72]. HRM has been shown to be a fast, reliable and cost-effective technique when characterizing a large number of samples likely to show low nucleotide variation [73].

The present work seeks to contribute to the knowledge of the northernmost known natural populations of *G. avellana* Mol. present in Chile. It corresponds to the first known report of the genetic structure and spatial distribution described for the species in the Region, which was developed mainly in natural populations that are not part of the National System of Protected Wilderness Areas of the State (SNASPE) administered by the National Forest Corporation (CONAF). At the same time, another of the objectives of this research is to propose an alternative methodology that may successfully replace more expensive molecular analyses or difficult interpretation, which could benefit conservation genetic studies using the analysis of HRM, which once optimized; is a fast, low-cost technique, especially when massive plant genotyping is required. Finally, the study aims at professionals interested in the conservation or ecological restoration of plants (e.g., Ecologists, Environmental Biologists, Botanists, and Molecular Biologists, among others) who wish to incorporate in their research alternative methodologies to their areas of specialization. When species of which little or nothing is known, it is advisable to use MaxEnt Model and HRM Analysis as a strategy for gathering rapid and integral information.

## 2. Results

### 2.1. Study Area and Plant Materials

One of the main challenges faced in this study was related to the limited knowledge of *G. avellana* distribution at the regional level. Therefore, it was relevant and transcendental for the successful development of the research to rely on the empirical and ancestral knowledge that we were able to record from the local peasant communities. These communities base part of their subsistence economy on the collection of fruits that they later market locally, either as dried or processed fruits (mainly as flour). As a result of this collaborative work and information provided by the locals and transmitted generationally, we were able to categorize geographically, phenologically, and phenotypically several centuries-old, natural populations of *G. avellana* not previously described that, for the most part, were part of peasants’ properties or other private sectors surrounding them. With this, we were able to select, evaluate and rescue trees using tissue culture techniques (unpublished data) with contrasting characteristics of interest and adapted to the edaphoclimatic zones of the region (Figure 1).

A total of twelve sites prospected in the Maule Region (Central Chile) where the presence of *G. avellana* was found, five of these belonging to National Reserves and seven private sites. In this study, the seven private sites were investigated, one site located in the Coastal Mountain Range and six sectors located in the Andes Mountain Range (Table 1). A total of 567 georeferenced points corresponding to adult trees were used to generate the distribution maps (Figure 2 and Figure 3). 

### 2.2. MaxEnt Model Projection of Potential Current and Future of G. avellana

To evaluate the suitability of the potential habitat of *G. avellana* in Central Chile (Maule Region) and project the possible variations that this habitat could suffer under different environmental scenarios, twenty-three different environmental variables were analyzed using the MaxEnt tool. Validation of our model allowed us to arrive at AUC values of 0.953 for the potential current habitat of *G. avellana*, indicating that the prediction results are very accurate, which can be used for subsequent analysis (Figure 4A). The range of potential habitats was “0” to “1”, while closest to “1”, the highest is the suitability (Figure 3B).

The results of the distribution model indicated that, in the Coastal range sector, there is a greater area of coverage of the potential current habitat suitability for *G. avellana* (Figure 3B). The Jackknife test indicates that the distribution of *G. avellana* was influenced by twelve bioclimatic variables (Figure 4B, Table 2). The bioclimatic variables that contribute the most to the model and to the changes in the distribution area correspond to Bio1 (annual mean temperature) and water steam (water vapor pressure), with 43.7% and 23.4%, respectively (Table 2).

Figure 5 shows that, under the different scenarios analyzed (CNRM-CM6-1 and MIROC-ES2L for four Shared Socioeconomic Pathways (SSPs): 126, 245, 370, and 585 (2021–2040; 2061–2080)), the populations of *G. avellana* naturally present in the Maule Region responded similarly to climate change in all these scenarios, where the potential future habitats were “0” to “0.5”, while closest to “0.5”, the highest is the suitability. This would indicate that *G. avellana* will continue to face serious risks a that pressure on the species will increase, mainly due to the expected increase in temperatures due to climate change, one of the key factors that determine the current potential distribution of *G. avellana* (Figure 4, Table 2). The prediction indicates that within the scope of 20 years (2040) and 40 years (2080) from now, the populations of *G. avellana* will see their ecological niche drastically diminished by 50% (AUC = 0.918), and their ideal habitat would move and concentrate in the highest sectors of the Andes mountains in height ranges that go from 520 to 1200 meters above sea level or higher.

### 2.3. Genotyping with High-Resolution Melting Analysis

The genotypes tested with eight EST-SSR markers and selected as HRM Standard or Reference Profile (Figure 13, Step 8) generated unique melting curves over different temperature ranges (Figure 14). The molecular markers evaluated subsequently in each population allowed the detection between three and five melting curves (Table 3). Figure 6 shows the profile melting and fragment analysis generated by the eight EST-SSR markers on a single genotype of *G. avellana*. Figure 7 shows the melting curves of seven representative genotypes of *G. avellana* from each population, using the EST-SSR marker Ga38. The shape of the melting curves revealed the difference between the genotypes and showed that all genotypes could be easily distinguished based on their melting curves.

Table 3 shows the diversity statistics for the eight EST-SSR markers used to assess the genetic diversity of 280 genotypes from the seven local populations of *G. avellana*. The optimal combination of the eight EST-SSR markers allowed the discrimination of all the individuals analyzed. On the other hand, the effective number of alleles per locus (Ne) for the total of the local populations was 1.40 ranging from 1.18 (Canelillo) to 1.55 (Punta de Águila). The coefficient of genetic differentiation (G_ST_ = 0.0431) indicates that 4.31% of the genetic diversity is found between populations and 95.69% within populations. The percentage of polymorphism in the analyzed populations remained in a range of 18.69%–55.14%, with the populations of Punta de Águila and Los Avellanos showing the highest percentages of polymorphism (55.14% and 49.53%, respectively). The situation that is maintained for these populations when analyzing Private alleles (Pa) whose values range between 59 for the Punta de Águila population and 53 for Los Avellanos.

The results of molecular variance (AMOVA) for the seven populations of *G. avellana* analyzed indicate that the highest percentage of variation is within the populations with 95%. followed by the genetic diversity present among the populations, with a total of 5% of the observed variance. Genetic differentiation between populations was moderate (F_ST_ = 0.052; *p* < 0.001) (Table 4).

The value of the Bayesian information criterion (BIC) based on the binary matrix (0 and 1) with the combined data of the eight EST-SSR markers showed that clusters useful to describe the data are K = 10 (Figure 8).

Figure 9 shows the selection criteria to determine the number of principal components chosen to plot the discriminant analysis of main components (DAPC) that explain the highest percentage of accumulated variation.

Despite the low levels of genetic differentiation, where the coefficient of genetic differentiation (G_ST_) in all populations fluctuated between 0.0339 and 0.0368 (Table 3), the DAPC analysis managed to identify 10 genetic populations and revealed that at least three of these genetic populations were clearly differentiated, while the other seven formed a cluster together (Figure 10 and Figure 11).

Figure 12 shows the distribution of the clusters. From the three clearly differentiated genetic populations (Cluster 2, 8, and 10) in the DAPC analysis (Figure 8), it can be determined that: (a) group 2 is made up of *G. avellana* genotypes from the natural populations of Roblería, Los Avellanos, Armerillo, and Punta de Águila; (b) Group 10 is made up of genotypes from Las Lomas, Roblería, Los Avellanos, Armerillo, and Punta de Águila; and (c) Group 8 is represented by genotypes from all populations.

## 3. Discussion

The current and future potential distribution of *G. avellana* described in the present work was developed by guiding us in the historical structure of ecological niche modeling reported in other publications [45,52,74], using for this the selection of independent variables that would allow us to obtain the final distribution map (Figure 3B), calibration based on data and evaluation of omission rates (AUC) (Figure 4A), Jackknife testing to see the contributions of the variables (Figure 4B), evaluation of the importance of the permutations (Table 2), and analysis of the variables against the response or adequacy curves (cloglog output) (Figure 4C).

Due to its simplicity, MaxEnt Model has become the main tool for quickly gathering information on ecological niches, since only two data input sources are needed to generate distribution models, the presence of data (georeferencing the presence of the species), and a set of environmental variables [75]. Regarding the data on the presence of *G. avellana* in the Maule Region, we decided to use only the georeferenced points of our fieldwork; one of the reasons for not using the data from the Global Biodiversity Information Facility GBIF is that this database incorporates records from different sources of information, museums, naturalists, and amateurs, which could lead to underestimating the information. In many cases, these records are based on observational data from amateurs who can make mistakes in classifying the species. Another drawback of GBIF is that records belonging to museum collections or places where the species has been introduced as an exotic species in a certain region can be incorporated; such is the case of *G. avellana*, which has been reported in the United States, New Zealand, and Peru.

A good model is given by the registration numbers; despite this, as a way of evaluating and calibrating our model, we decided to discard some records where the species is currently present (National Reserves). We were able to show that our model was adequate, because once the distribution map was generated (Figure 3B), it accurately included the areas not considered within the dataset. In this regard, it should be noted that, in the Maule Region, there are seven national reserves, part of the National System of Protected Wilderness Areas of the State (SNASPE) (R.N. Radal Siete Tazas, R.N. Los Ruiles, R.N. Los Queules, R.N. Laguna Torca, RN Federico Albert, RN Los Bellotos del Melado, and RN Altos de Lircay). Here, only in five of the reserves, the presence of *G. avellana* could be evidenced, areas that coincide with the populations not considered in the MaxEnt modeling but that were nevertheless successfully extrapolated in the representation of the projections obtained from the final model (R.N. Los Ruiles (Coastal area), R.N. Los Queules (Coastal area), R.N. Altos de Lircay (Mountain Andes), R.N. Radal Siete Tazas (Mountain Andes), and R.N. The Bellotos del Melado (Mountain Andes)).

We can obtain relevant information when analyzing the response curves of our variables (Figure 4C), where the limit of our accessible area is on the x-axis (variables), and the response or adequacy (cloglog output) is on the y-axis. We seek that our adequacy is continuous. Therefore, if the curve goes up, extrapolating will be very risky, since, as the variable increases, it will also increase the response or adequacy and vice versa in descending curves; there will also be the risk of extrapolation, and this occurs especially when dependent variables are included in the model.

Therefore, when performing the test calibration of models created by default in MaxEnt for *G. avellana* with different registry data and all the bioclimatic variables, all the possible predictive models yielded AUC values greater than 0.5 and where the distribution maps were very similar. This means that all the models were better than chance and that they also have a very good way of controlling themselves, possibly because the presence of *G. avellana* is limited to different environments (Coastal Mountain Range and Andes Mountain). At this point, various authors recommend performing a preselection of candidate covariates before modeling in order to improve and increase the accuracy of the predictive model by using the Pearson correlation coefficient or multiple regression [49].

MaxEnt reduces the effect of one variable with another that is correlated by bootstrap resampling where truly independent variables are expected to be present in the largest number of bootstrap samples; Thus, in the case of the final model described for *G. avellana*, the twenty-three variables MaxEnt reduced to twelve variables that are not correlated (Figure 4C, Table 2). It should be noted that MaxEnt is a tool that predicts and projects suitable distribution areas of the species; however, when carrying out an in-depth analysis of the projection of the obtained final model, other aspects that are not related to the bioclimatic variables analyzed must necessarily be considered. For example, in our current distribution model in *G. avellana* in the Maule Region, the suitable areas were categorized on a color scale ranging from 0 to 1, in different sections 0.00, 0.33, 0.65, and 1 (Figure 3B), with values that MaxEnt also alternatively categorizes regrouped into four classes: unsuitable (0–0.23), low potential (0.31–0.38), moderate potential (0.46–0.77), and high potential (0.85–1.0). According to this analysis, and based on the analyzed variables, in the coastal sector, *G. avellana* should be found in a wide range of geographical distributions (our field data confirm this distribution). However, the amplitude of the areas projected on the map tends to indicate that there is a high probability of finding a high number of populations, which does not adjust to reality, because *G. avellana* in these areas is represented by few individuals and confined to small forest native fragments. The question we asked ourselves as researchers was, is there an adjustment error when calibrating the model? There could be an alternative; nevertheless, a point to consider in the interpretation of the results is that the model is based on bioclimatic data and does not consider human activities. In recent years, in these coastal areas, the distribution of *G. avellana* has been affected by the high level of human settlement, which is higher than what occurs in high mountains.

In case of the evaluation of climate change, the value of these bioclimatic variables is fundamental for the interpretation of the effects of a current scenario against a change in the future scenario (Figure 5). Both the global climate models and the Shared Socioeconomic Pathways used to assess the potential future distribution of *G. avellana* can produce different maps concerning changes in response or suitability, and they can also produce slightly different areas for where one is extrapolating, i.e., when the models are compared, adjustments or mismatches can occur depending on which variables are used.

Using the usual parameters described in MaxEnt to analyze the ecological niches of a species whose presence is reduced to specific environmental areas, as is the case of *G. avellana*, provides quite controlled models; however, when the distribution of a species in specific environmental areas is rather uniform (Wallacean species), where we know that the distribution of a species is in almost all accessible areas, finding a model that identifies well where the species is and where it is less suitable becomes quite complex.

Currently, the idea is proposed that there is no best model; there may be several best models, and it is necessary to have different parameters to evaluate them (a set of different variables, multipliers, and response types) [76]. In this publication, the authors delve into ecological niche modeling using MaxEnt and present the Kuenm, which is an R package created to improve the way we are creating ecological niche models using MaxEnt, to achieve a better interpretation of the results obtained. Kuenm is in charge of model calibration, construction of final models, extrapolation of the risk analysis, and interpretation of the results obtained from the final models, especially when we make projections. In MaxEnt, there are seven types of responses and 29 combinations, and each one of them has a different implication. What is the best combination? According to the author, it is probably not only one; therefore, it is a combination that can be defined in the following equation: 10 groups of variables x 15 multipliers x 7 response types = 1050 candidate models to evaluate. The proposal is that all the models consulted better than chance, only those with omission rates lower than a pre-established threshold (5 or 10%) are considered, and among the latter, those that are sufficiently simple; that is, that the model detects adequacy in areas so close to the environmental zones where the records are. Kuenm allows consensus on the application of the statistics of all these models, it allows these statistics to be obtained through parameters (mean, median, variance, and standard error), together with a representation of changes in the appropriate areas at different times (climate change scenarios). However, despite the many advantages described for Kuenm, it still requires certain adjustments for its massive use, such as improvements in the calibration process (parallelizing functions) and optimizing the tools for data optimization to start the analysis, among others.

Regarding the genetic analysis of *G. avellana*, we decided to follow the strategy described in Figure 13, because our knowledge of the species in the region was practically null; we began by prospecting a field-level survey to find the study populations; subsequently, we selected a group of individuals under phenotypic, phenological, and edaphoclimatic criteria that presented contrasting characteristics and of interest for our research (Figure 13, a). The strategy designed to apply the HRM analysis in *G. avellana* is summarized in nine steps described in Figure 13. In terms of describing the genetic variability of this small group of selected individuals, we use a type of molecular marker, whose applications are widely reported, the so-called Nonanchored Inter Simple Sequence Repeat (ISSR) markers [77,78,79]. These markers are of the dominant type, and among their many advantages is that it does not require prior knowledge of the genome sequence of the organism under study. The genetic profiles can be visualized in gels of agarose and are easy to interpret, since the characteristic PCR pattern is considered the “genetic fingerprint” of each of the individuals analyzed, where the presence of a band represents the dominant genotype (homozygous or heterozygous). ISSR markers detect high variations, are highly reproducible, and are inexpensive. The objective of performing this previous stage was to find and develop our internal controls (HRM Standard) to be used in the massive genotyping of *G. avellana* using HRM analysis. With the use of these markers, a low genetic variability was determined among the selected individuals (data not shown). Subsequently, the individuals that presented variants in their genetic profiles were selected as possible candidates to optimize the HRM methodology and also be used as reference genotypes (Figure 13, Step 3–Step 9).

It should be noted that the methodology described was designed for a native species where genetic knowledge in the region was almost null. Therefore, this methodology can be used as a reference for other native species of interest whose knowledge is also limited. According to our criteria, the key point to subsequently use the HRM analysis is the optimization of the PCR in any of its steps (Figure 3, Step 2, Step 4, and Step 7) that allow the selection of reference genotypes and adequate genetic markers in such a way to obtain clear and reproducible HRM profiles for massive and accurate plant genotyping. This methodology can be adjusted, and some steps can be discarded, considering other factors, such as the required application, the needs of the investigation, or the previous knowledge of the species under study, for example: if it is required to evaluate the variation nucleotide of a particular gene, specific probes can be designed, and the usual optimization of HRM analysis can be performed. Another alternative might be if the researcher has defined their markers and controls; the analyzer will only need to follow the optimization described in Figure 13, Step 7–Step 9.

Regarding the optimization of the parameters of the HRM analysis in *G. avellana*, even though the EST-SSR molecular markers are of the codominant type, in this study, they were used as dominant markers, as shown in Figure 14. Each marker used has several different reference genotypes with which the individuals of each population are compared to subsequently generate the binary data matrix. Considering only the shape of the melting curve and analyzing the profiles generated jointly by the 8 EST-SSR markers on each population of *G. avellana*, few HRM profiles are described (Table 3). This agrees with the moderate Genetic differentiation found between populations (Fst = 0.052; *p* < 0.001) (Table 4). These data are consistent with those reported for the populations analyzed in Southern Chile, where the analysis of molecular variance (AMOVA) showed a moderate genetic differentiation (Fst = 0.118; *p* < 0.001), also showing that the greatest genetic diversity is found within the populations (89.07%; *p* < 0.001) [80]. This low genetic diversity was also reported by other researchers who evaluated the genetic diversity of *G. avellana* populations in other regions in the south of the country through the use of different molecular markers, including AFLP, and cpDNA, among others [81,82]. In other words, if we analyze Figure 14, the profiles on the left show HRM standard profiles (reference profile) of genotypes of *G. avellana* Mol. analyzed using two EST-SSR markers (Ga36 and Ga88), and seven melting curves are observed, the number of which will depend on the number of reference genotypes and markers analyzed, as these curves do not vary too much in their shape; we could say that there are only three HRM profiles for these two markers (simple downward curve, simple upward curve, and double curve). Now, if it is considered that an HRM Profile is made up of both the shape of the melting curve and the dissociation temperature, this number increases (Table 3, 28 HRM profiles); this is how they should be considered when building the binary matrix. For example, if we observe, in Figure 14, images b (80.3 °C) and d (81 °C), their shapes are similar, but the dissociation temperature is different when comparing the melting curve of individual 1 of population 1; it has a similar curve shape for both reference genotypes (b and d), but its melting temperature is similar (81 °C) to the reference profile of image d (81 °C). This leads us to interpret the possible reason why the greatest genetic diversity is found within the populations (Table 4) [80].

The advantage of HRM analysis is that Real-Time PCR system software provides a table of dissociation temperature data automatically (including clustering graphs), and for the construction of the binary matrix, only must be analyzed the forms of melting curves, which makes the process something fast, precise, easy to interpret and analyze, and above all, an independent process, since high-cost external services are not required to obtain results. Even more, when it is required to analyze small fragments (Figure 6 and Figure 7), it allows the replacement of cumbersome or expensive methodologies such as polyacrylamide gels or the use of capillary electrophoresis.

Our experience also led us to give another application to HRM analysis, which consists of determining the genetic stability of *G. avellana* plants propagated by in vitro culture (Figure 13, Step 3, a). It is known that plants regenerated from tissue culture are prone to genetic alterations due to stress induced by in vitro culture conditions and regeneration mode and that these somaclonal variations can be detected through the use of molecular markers; thus, the clones of the *G. avellana* genotypes propagated for several generations were evaluated with the HRM analysis with their respective genotypes of original called “mother plants” (Figure 3, Step 1, a). Based on our HRM methodology optimized for *G. avellana*, we were able to determine that the methodology used for plant propagation did not generate mutations in the clones of the propagated genotypes.

Regarding the Discriminant Analysis of Principal Components (DAPC), it allows us to know genetic clusters that can be biologically meaningful structures and reflect interesting biological processes. For this, we must define the number of clusters that are useful to describe our data (Figure 8 and Figure 9). DAPC can benefit from not using too many PCs. Indeed, retaining too many components concerning the number of individuals can lead to overfitting and instability in the membership probabilities returned by the method [83]. Under this aspect, we were able to determine that the number of minimum conglomerates that describe over 80% of these 10 genetic groups (genetic cluster) was 40 PCs (Figure 8, Figure 9 and Figure 10) and that this grouping remains very stable when increasing an 80 PCs; in which case, 100% of the accumulated variance is explained (Figure 11). Therefore, *G. avellana* presents seven geographic groups and 10 genetic groups; considering that natural populations of private enclosures were evaluated, this could be indicating that the populations may not be completely natural and that there is a genetic flow, perhaps due to the introduction of material from other localities, as has been the case with the projects for the establishment of *G. avellana* in the region where they were used seeds from other areas in the south of the country [1,8].

Based on the Model MaxEnt results and HRM Analysis, we could make some decisions in terms of conservation. For example, if we aim to protect populations containing significant genetic variability (Table 3, Figure 12) in areas where the climate would have less effect (Figure 5). By complementing this information, the priority populations for conservation would correspond to the areas of Los Avellanos and Punta de Aguila, because these populations are adapted to the edaphoclimatic conditions that will present less disturbance in the face of climate change (Figure 2B and Figure 5). In addition, they contain a high allelic richness (Table 3) and a high representation of individuals in the ten genetic groups obtained with the HRM analysis and DAPC (Figure 12). Now, if our purpose is to protect populations in areas threatened by climate change and that also contain more numerous representations of a particular genetic group, the priority population to conserve would correspond to the population of Canelillos (Table 3 and Figure 5 and Figure 12). Integrating the MaxEnt and HRM analysis as a whole generated rapid, accurate, and transferable information on the current status of the *G. avellana* populations present in the Maule Region, information that will allow the local community and government entities to make conscientious, better informed and objective decisions in a timely manner when implementing strategies aimed at guaranteeing effective plans for the conservation, recovery, make visible, and potential use of the species.

Native plants are often promoted to solve various environmental sustainability issues [84]. Promoting the conservation of native species can generate a series of benefits, either from an economic (e.g., development of marketable forest and agricultural products), social (e.g., improvement of unemployment and local poverty rates), or environmental point of view (e.g., conserving biodiversity and improvement of soil and water quality) [85]. Modern science has made it technically possible to “domesticate” species undomesticable in the past [86]. In degraded ecosystems, evolution can be assisted by considering the inclusion of plants that reflect general evolutionary patterns, that can adapt to the selection of modified environments, and that contribute to the restoration of ecosystem structure and function [87]. In this context, *G. avellana*, due to its ethnobotanical characteristics, is a species with a high potential for domestication [88]. The experiences developed in Chile indicate that it is feasible to establish the species for timber and fruit purposes if the appropriate techniques are applied to the specificities of the species [89], though the low percentage of hazelnut survival (less than 35%) is pointed out, possibly associated with different strategies in the acquisition of nutrients [90]. Several studies have raised the need to implement genetic improvement programs for the species by selecting individuals with specific characteristics based on the genetic variability of the species and edaphoclimatic plasticity in such a way to have collections with ecotypes of high productive adaptability [14]. Currently, the use of *G. avellana* is restricted to the collection of its fruits in their natural state; however, less than thirty percent of the total inventory is considered suitable for collection, due to the difficult access to the populations and their high dispersion space [91]. Thus, its exploitation by harvesting in its natural state is limited and is not enough to guarantee the demand of the industry dedicated to the processing of fruits, due to the fluctuation that exists in its productivity and the disappearance of the native forest due to natural factors and anthropic activities [92].

Currently, climate change is becoming a serious challenge around the world, with significant threats to food security, ecosystems, economic stability, and water resources [93]. The impact of elevated atmospheric eCO_2_ and associated shifts in temperature and precipitation are all expected to impact plant ecophysiology, distribution, and interactions with other organisms [94]. Accordingly, concerns over species extinction are warranted, as it provides food for all life forms and primary health care for more than 60–80% of humans globally [95]. The change in global climate is due to the increasing concentration of greenhouse gases (GHG) in the atmosphere, the increasing global temperature over the past century by about 0.8 °C and is expected to rise between 0.9 and 3.5 °C by 2100 [96]. To mitigate the negative impacts of climate change, it is recommended to restore degraded sites, and develop sustainable forest management and community-based biodiversity conservation [95]. A recommended strategy for this purpose is to include peasant communities and indigenous peoples in these reforestation plans, promoting the concept of “natural regeneration managed by farmers”, a type of productive agroforestry that adds value to the units of conservation, considering economic benefits for both restoration and carbon programs, as well as for improving the quality of life and local economy of peasants and their families through the commercialization of timber and non-timber forest products [85].

Based on the above, our focus of attention was focused on making visible a phytogenetic resource that is almost unknown at the regional level and with great potential for domestication and exploitation, such as *G. avellana*, generating the greatest amount of information that could be quickly socialized and transferable in such a way as to encourage and invite the scientific community, farmers, local community, and government agencies to develop and accelerate conservation strategies and ecologically sustainable use of the species. Our biotechnological approach allowed us to contribute significantly to one of the key and fundamental questions in this type of study related to what and where to restore. The criteria that we address to develop an effective plan for conversation and/or sustainable use in *G. avellana* not only considered a survey of genetic and ecological information of the species in the region but also included the development of a fast and efficient system of in vitro propagation of individuals of high genetic and phytosanitary quality selected based on their genetic, phenotypic, phenological plasticity, and agroforestry characteristics of productive interest (unpublished data). This point is really important since Chile has committed to restoring 600.000 ha of native forest by the year 2035; however, the quantity and quality of native plants that are required to be produced were not considered; When evaluating whether the nurseries currently meet the minimum seedling quality standards based on their morphophysiological attributes, it was concluded that the restoration commitments desired by 2035 would not be achieved until 2181 [97]. With respect to the potential for domestication, *G. avellana*, due to its agroforestry and productive characteristics, should be seriously considered as an alternative for the development of a crop of regional interest. If we consider, for example, that the commercial plantations in the Maule Region of its homonymous European Hazelnut (*Corylus avellana* L.) have increased their representation significantly, going from 30 ha in 1999 to 24.455 ha in 2021, with a representation of this species at the country level by over 80%, becoming, in addition, the fourth most important fruit crop at the country level [98], thinking of *G. avellana* as an ecological crop, more environmentally friendly, and with regional identity is a completely viable alternative to implement. However, at this point, we consider that the social, cultural, ecological, and environmental focus should not be lost; therefore, we propose not to think of *G. avellana* for the development of a massive monoculture but rather the development of a “domestication limited to small-scale” fully regulated (through Technology Transfer), which allows both the conservation of genetic diversity and the sustainable use of the species and specifically contributes to improving the local communities’ quality of life. These communities are made up mostly of peasants whose families own small extensions of land that they have inherited generationally, where there are remnants of native forests—that is, fragmented forests of native species, including *G. avellana*. Lands that, due to the edaphoclimatic and topographical conditions of the area, are generally not suitable for the establishment of other types of crops that allow the development of agricultural activities. These lands have soils with low water availability, often eroded, with streams that are difficult to access, where, in addition, in the case of *G. avellana*, although there are natural populations within these lands, the production of its fruits is uneven and insufficient and makes it unfeasible to base the family economy on a permanent and profitable activity. Under this social and cultural reality, and as there are no incentives from the state towards these communities to protect and maintain these native resources, the peasants and locals who own these lands dedicate themselves to working in other economic activities, and with regard to the management of the use of their land, they often make decisions to sell part of it (invoking the change in land use regulations) or they simply choose to cut down the remnants of native forest to replace them with forest plantations and more economically lucrative fruit trees such as Pine (*Pinus radiata*), Eucalyptus (*Eucaliptus globulus*) and, in recent years, European hazelnut (*Corylus avellana*). According to the above, and with the information generated in this study, it is feasible to develop an adequate platform for multipurpose technology transfer in *G. avellana* and transform it into an adequate alternative for reforestation, since its benefits and scope in different areas (ecological, environmental, social, and cultural, among others) have not been really valued or dimensioned.

## 4. Materials and Methods

### 4.1. Study Area and Plant Materials

The plant materials used in this study were obtained from The Maule Region (VII Region, Central Chile), located between 34°41′ and 36° 33′ south latitude (Figure 2A). Young leaves, fruits, and plant explants were collected from seven natural populations of *Gevuina avellana* Mol. located in two Andean sectors of the Maule Region (Andes and Coastal Mountains) (Figure 2B, Table 1). These populations consisted mainly of adult trees from 70 to more than 100 years old (cultural record of local communities). We prospected around 700 trees, of which only 280 were used for genetic analysis, selected based on their adaptation to extreme edaphoclimatic zones and agronomic rate characteristics (for example, age, fruit size, and high fruit production year to year).

### 4.2. MaxEnt Model Projection of Potential Current and Future of G. avellana

#### 4.2.1. Data Sources

The current known distribution data of *G. avellana* in Chile was obtained from Global Biodiversity Information Facility (GBIF, https://www.gif.org, accessed on 22 June 2022) (Figure 3A). The current distribution of *G. avellana* in Maule Region was recorded during our field work; we georeferenced a total of 567 distribution points of *G. avellana* in the Region Maule with Professional Garmin Gpsmap 65.

#### 4.2.2. Variables Selection

Twenty-three variables were retrieved as predictors to model the potential environmental niche of *G. avellana* based on its current presence dataset. In particular, 22 bioclimatic layers and one topographic variable (elevation) were obtained from the WorldClim database (https://www.worldclim.org, accessed on 22 June 2022). Variables were retrieved at a spatial resolution of 30 arc-seconds (~1 km) for the current distribution and 2.5 arc-minutes (~21 Km) for future distributions. Projections of future climate change to predict the potential distribution of *G. avellana* were evaluated using the so-called Scenario Model Intercomparison Project (ScenarioMIP) under Phase 6 of the Coupled Model Intercomparison Project (CMIP6). Two global climate models (GCMs), CNRM-CM6-1 and MIROC-ES2L for the four shared socioeconomic pathways (SSPs), 126, 245, 370, and 585 (2021–2040; 2061–2080) were evaluated.

#### 4.2.3. Data Analysis

The georeferenced longitude and latitude variables were analyzed with QGIS version 3.20 software and duplicate data were eliminated. Geographic location map was made through the Geographical Information System (GIS) using tools ArcGIS (ArcMap 10.03 software). The coordinates were exported to CVS format and the climatic variables in ASCII format, both integrated into MaxEnt software version 3.4.4 for further analysis [99]. First, the complete model with all the variables. Finally, the twelve most influential variables (Radiation, Water Steam, Wind, Bio1, Bio2, Bio3, Bio4, Bio6, Bio13, Bio15, Bio 16, and Bio17) were retained in the process in MaxEnt (Figure 4B).

### 4.3. Genotyping with High-Resolution Melting Analysis

#### 4.3.1. Nucleic Acid Preparation

The genomic DNA was extracted from 500 mg of lyophilized leaves of Chilean hazelnut according to the protocol CTAB described by [100]. The leaves were pulverized in a MM 400 biological sample disruptor equipment (www.retsch.com (accessed on 20 September 2022)). To carry out the exploratory analysis of the genetic variability of the populations, genomic DNA was extracted from 60 trees from different natural populations of *G. avellana*. They were selected based on characteristics of interest (Figure 13, Step 1, b). Subsequently, the individuals that presented differences in their genetic profiles were introduced into the in vitro culture, and a genomic DNA extraction was performed on the propagated explants (Figure 13, Step 3, b) to select the genotypes with nucleotide variants to be used as HRM Standard (Figure 13, Step 6 and Step 8, b). Finally, genomic DNA was extracted from 280 trees from the seven study populations (Figure 13, Step 9, a) for the genotyping of *G. avellana* populations.

The nucleic acids purity and concentration were estimated on a Multiskan Sky High^TM^ (Thermo Fisher) and displayed in 1% agarose gel electrophoresis. The final concentration was adjusted to 10 ng µL^−1^.

#### 4.3.2. Selection of Molecular Markers

For the exploratory analysis of the genetic variability of individuals of interest, the so-called Nonanchored Inter Simple Sequence Repeat (ISSR) markers were used [77]. Different molecular markers have been deployed to assess the genetic diversity among populations of *G. avellana* [19,81,82]. We selected EST-SSR Molecular Markers specific for *G. Avellana,* because their characteristics were adequate for analysis HRM [80].

#### 4.3.3. Standardization and Selection of HRM Standard

The strategy designed to apply the HRM analysis in *G. avellana* is summarized in nine steps described in Figure 13**.** This consisted of selecting adult trees from the seven populations under study with contrasting phenological and phenotypic characteristics of different kinds (differences in flowering time, tolerance to stress and fruit size) and introduce explants of each one of them through tissue culture techniques (in vitro culture) that would allow having a core collection representative of the diversity found. That is, DNA sample variants were isolated and conventional PCR conditions were standardized on these genotypes with expressed sequence tag-simple sequence repeat (EST-SSR) markers described for the species [80]. Of a total of 20 molecular markers analyzed, eight were selected for genetic analysis due to their high reproducibility. Subsequently, the PCR fragments obtained were visualized in a 2% agarose gel, and their sizes were determined with the GelAnalyzer 19.1 software. The fragments obtained for the different genotypes and analyzed with eight EST-SSR that showed differences in size were subsequently sequenced at the company Macrogen (Seoul, Korea).

The standardization of the HRM profiles and selection of the HRM Standard was carried out by measuring the PCR kinetics of the EvaGreen fluorophore in real-time equipment and was performed based on the order of the typical experimental workflow described in a practical guidance for high-resolution melting (HRM) [101]. For the HRM Standard selection, the *G. avellana* genotypes that presented variations in their nucleotide sequences were selected and evaluated by HRM. To assess the reproducibility of the HRM method, PCR reactions were performed in triplicate for each genotype for the eight EST-SSR primers (Figure 13, Step 8). In this way, the genotypes that presented clear differences in shape, and melting temperature (HRM profiles), were used as HRM Standard for the genotyping of *G. avellana* populations (Figure 13, Step 9).

#### 4.3.4. Genotyping of Populations of *G. avellana* with High-Resolution Melting Analysis

The populations of *G. avellana* Mol. were genotyped with eight expressed sequence tag-simple sequence repeat (EST-SSR) markers, based on a cDNA library of the species: Ga94, Ga38, Ga88, Ga90, Ga36, Ga7d, Ga92, and Ga49 [80]. Nucleic acid extracts 10 ng µL^−1^ (1 µL of each) were used directly in 10 µL reaction volumes containing 1X Kapa HRM Fast Master Kit (Kapa biosystems, Cape Town, South Africa), 2.5 mM MgCl_2,_ and 200 nM of each primer. PCR amplification, DNA melting, and end point fluorescence level acquiring PCR amplifications were performed on AriaMx Real-Time PCR System (Agilent Technologies, Inc. 5301 Stevens Creek Blvd Santa Clara, CA 95051 USA). PCR protocol was established for all SSRs, under the following conditions: 40 cycles of denaturation at 95 °C for 10 s, annealing at 60 °C for 30 s, and extension at 72 °C for 20 s. High-resolution melting analysis was performed after PCR amplification at the temperature ramping and fluorescence acquisition setting recommended by the manufacturer, which is, the temperature ramping up to 95 °C and rising by 0.2 °C/10 s.

#### 4.3.5. Data Analysis

Two hundred and eighty trees of *G. avellana* from seven natural populations of the Maule Region were analyzed. HRM profiles of each individual were compared to representative standard curves for each marker (HRM Standard) (Figure 14). One Binary matrix (280 rows × 107 columns) were generated where “1” indicated similarity over 90% with respect to the curve, and a melting temperature of the HRM Standard and “0” indicated the absence of similarity. The binary matrix of 0 and 1 generated with the integrated data of the eight EST-SSR markers were used for genetic analyses using different programs, such as POPGENE 1.3.2 (Table 3) GenAlex 6.5 (Table 4) and Adegenet package for R to generated Discriminant Analysis of Principal Components (DAPC) (Figure 8 and Figure 9). DAPC is a multivariate method designed to identify and describe clusters of genetically related individuals. When group priors are lacking, DAPC uses sequential *K-means* and model selection to infer genetic clusters. DAPC was developed following tutorial described by Jombart [83]. Using general parameters called find.clusters to identify clusters and dapc to describe the relationships between the clusters. To do this, we began by downloading the *Adegenet* package and ade4 software (>library adegenet and ade4). Then, we established a directory with the data using the following parameters: (1) dat <- read.table (binary data matrix in TXT format), (2) obj <- genind to define how our data is composed (ploidy, number of individuals and populations, type of matrix), (3) parameter “grp<-find.clusters (max.n.clust = 40)” to define and form the population groups, and (4) parameter “dapc1<-dapc (obj, grp$grp)” were used to choose the PCs, number of clusters, and the number of eigenvalues that explained the highest percentage of the accumulated variance”, which finally allowed obtaining the basic scatterplots using function (5) “scatter (dapc1)”.

## 5. Conclusions

It was shown that EST-SSR genotyping in *G. avellana* with HRM analysis can be a fast and cost-effective tool, suitable to replace other genotyping technologies such as DNA sequencing by capillary electrophoresis. We conclude that the HRM methodology described in this work can be used as a reference and applied to the study of other native species of interest.

Based on the ecological niche modeling developed by MaxEnt, we determined that currently the ideal areas in the Maule Region to establish *G. avellana* Mol. They are located in the Andean sectors of the Coastal Range and Los Andes range, areas that, by the year 2040, will decrease by 50%, moving mainly to the Andes range above 1200 meters above sea level. Along with this, according to analysis HRM and the discriminant analysis of principal components (DAPC) we conclude that high-priority areas for protection correspond to Los Avellanos and Punta de Águila populations due to their greater genetic diversity and allelic richness.

According to the valuable interaction that allowed us to establish a synergy between the real needs of the peasant community dedicated to the activity of collection of fruits in *G. avellana*, and the interest shown in wanting to be part of a solution to a problem that affects us all as a society related to negative anthropogenic impacts and climate change on our plant genetic resources, we conclude that the key for a successful implementation of a conservation program in *G. avellana*, is to actively include the participation of the peasant community to maintain this synergy between them and the scientific world.

## Figures and Tables

**Figure 1 plants-11-02803-f001:**
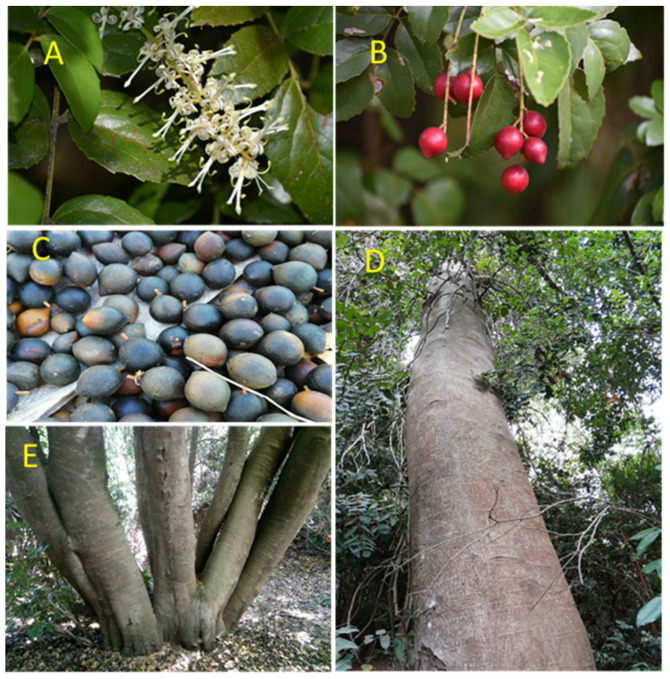
*Gevuina avellana* Mol. (**A**) Inflorescence. (**B**) Immature fruits. (**C**) Ripe fruits. (**D**) Straight trunk. (**E**) Branched trunk.

**Figure 2 plants-11-02803-f002:**
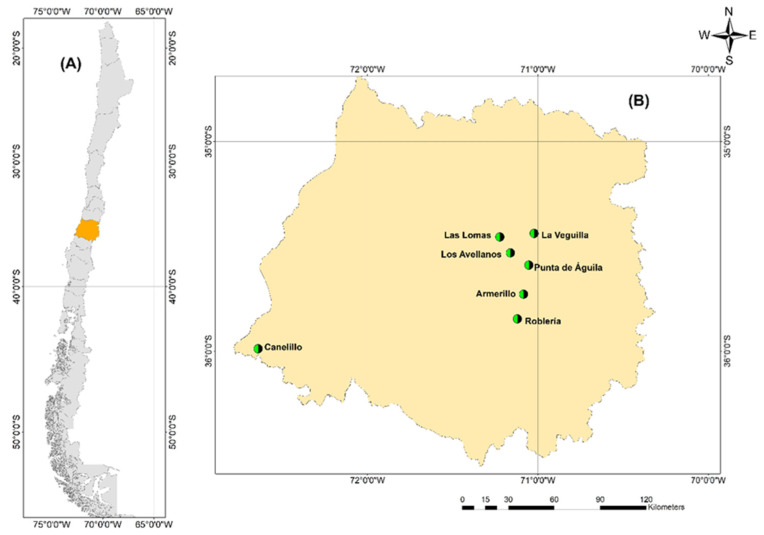
Maps showing geographical locations where samples were collected. (**A**) Study area in Chile named the Region of Maule (orange area on the map). (**B**) The marked points are exact points of collection sites of the seven populations of *Gevuina avellana* Mol.

**Figure 3 plants-11-02803-f003:**
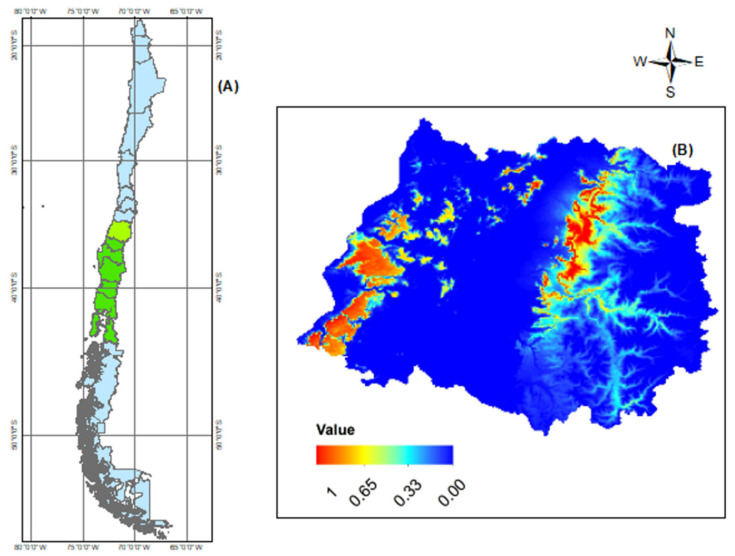
Maps of distribution of *G. avellana.* (**A**) Current distribution in Chile (Green area) and (**B**) potential current habitat suitability in Maule Region; while closest to “1”, the highest is the suitability for *G. avellana*.

**Figure 4 plants-11-02803-f004:**
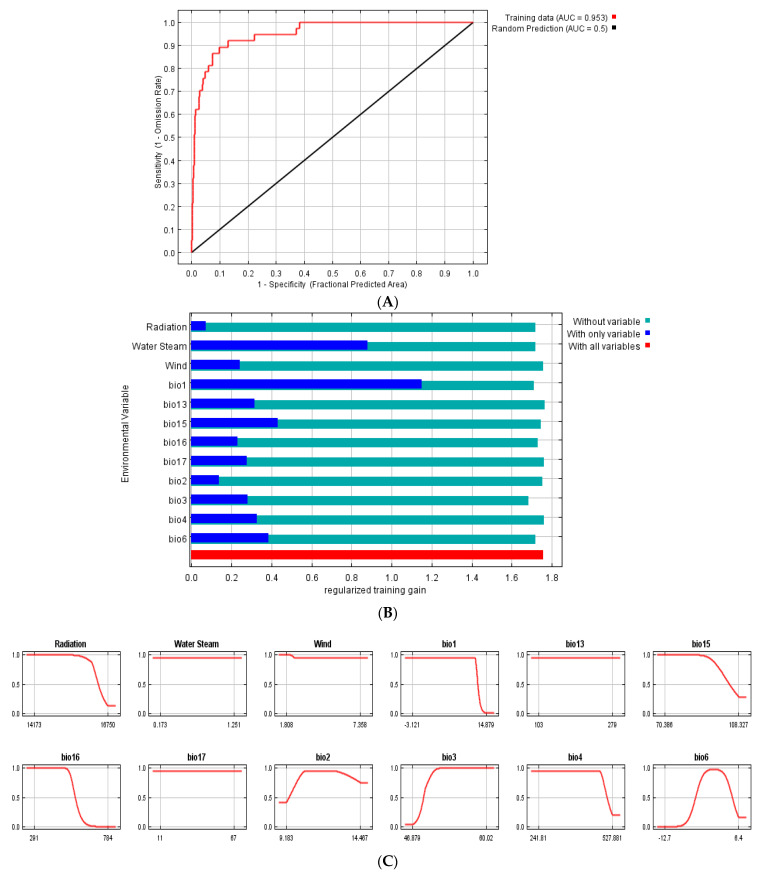
Maxent model current projection for *G. avellana.* (**A**) Analysis of omission/commission; pictured is the receiver operating characteristic (ROC) curve. (**B**) Picture shows the results of the permutation test of variable importance (for codes detail, see Table 2). (**C**) Response curves of *G. avellana* to variables.

**Figure 5 plants-11-02803-f005:**
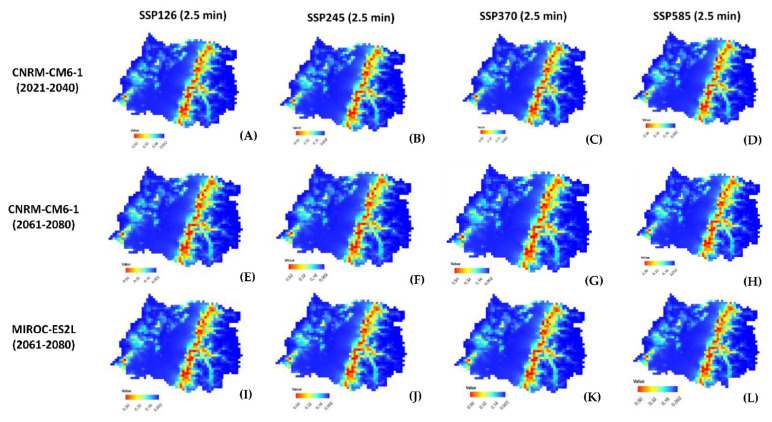
Ecological niche modeling of *G. avellana* in Maule Region based on the predicted climate change of two global climate models (GCMs): CNRM-CM6-1 and MIROC-ES2L and four Shared Socioeconomic Pathways (SSPs): 126, 245, 370, and 585 over 20 year periods (2021–2040; 2061–2080), while closest to “0.5”, the highest is the suitability for *G. avellana* (AUC = 0.918).

**Figure 6 plants-11-02803-f006:**
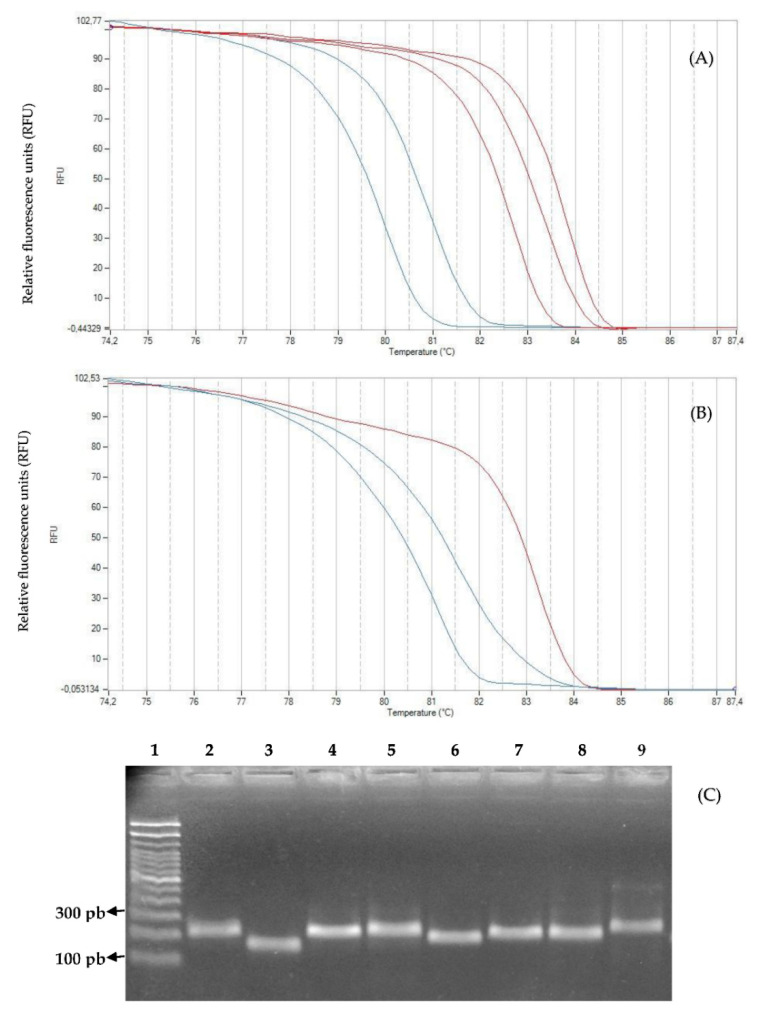
HRM profile of unique genotype of *G. avellana* Mol. (code 43C01) using eight EST-SSR markers (displayed graph from left to right) (**A**) Genotype analyzed with the markers Ga94, Ga38, Ga88, Ga90, and Ga36. (**B**) Genotype analyzed with the markers Ga7d, Ga92, and Ga49. (**C**) Fragment analysis of unique genotype of *G. avellana* Mol. using eight EST-SSR markers in agarose gel electrophoresis (2%). Lane 1, 100 bp DNA Ladder (Invitrogen 15628019); Lane 2, marker Ga7d (224 pb); Lane 3, marker Ga36 (180 pb); Lane 4, marker Ga49 (223 pb); Lane 5, marker Ga90 (228 pb); Lane 6, marker Ga94 (203 pb); Lane 7, marker Ga38 (219 pb); Lane 8, marker Ga88 (218 pb); Lane 9, marker Ga92 (242 pb). Genomic DNA of the *G. avellana* genotype code 43C01 was used.

**Figure 7 plants-11-02803-f007:**
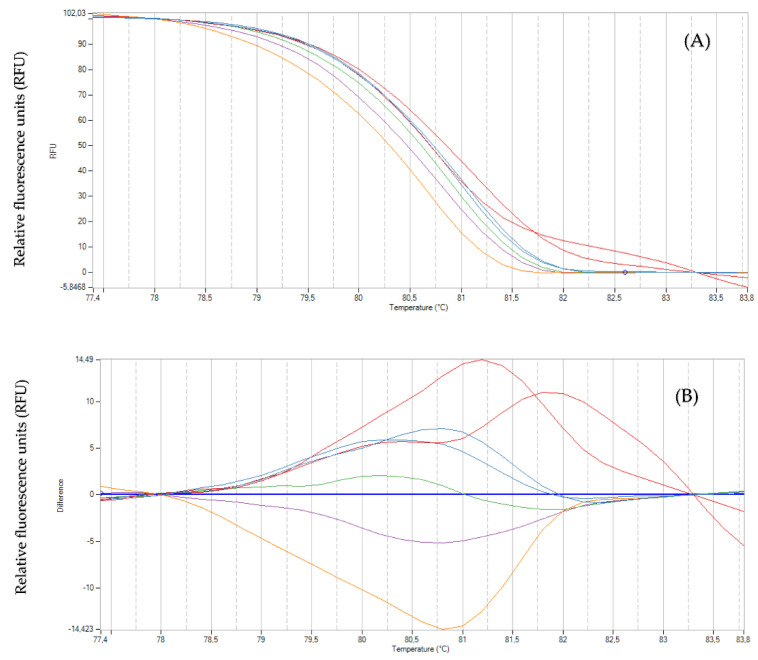
Discrimination of seven unique genotypes of *G. avellana* using microsatellite marker Ga38. (**A**) Normalized HRM graph of seven unique genotypes of each population of *G. avellana* Mol. (**B**) Difference plot of those seven unique genotypes, HRM melting curves (displayed graph from top to bottom) Armerillo (Red), Punta de Águila (Burgandy), La Veguilla (Sky-blue), Los Avellanos (Light blue), Las Lomas (Green), Canelillo (Purple), and Roblería (Yellow).

**Figure 8 plants-11-02803-f008:**
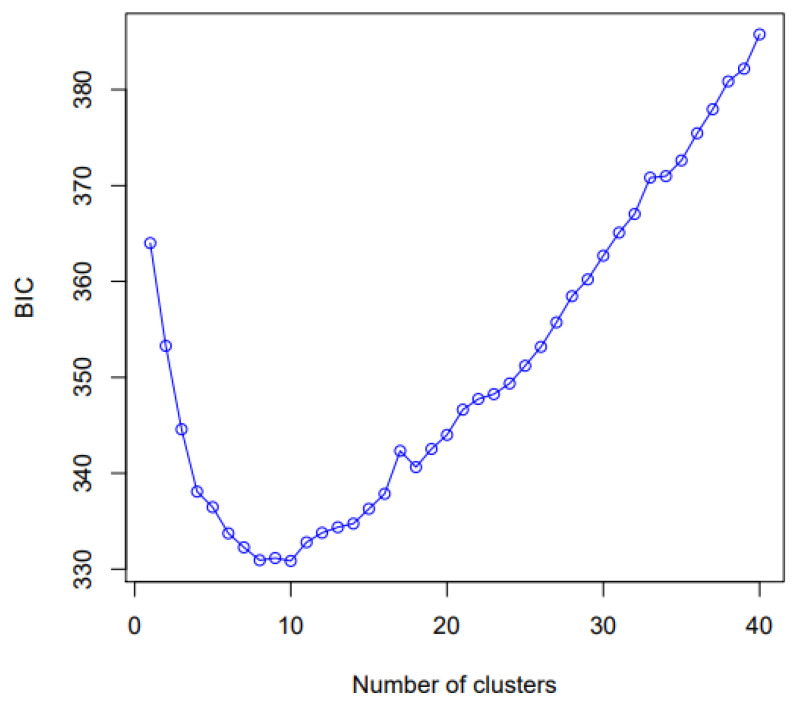
Value of Bayesian information criterion (BIC) versus number of clusters. This graph shows a clear decrease of BIC until *k* = 10 clusters; after which, BIC increases.

**Figure 9 plants-11-02803-f009:**
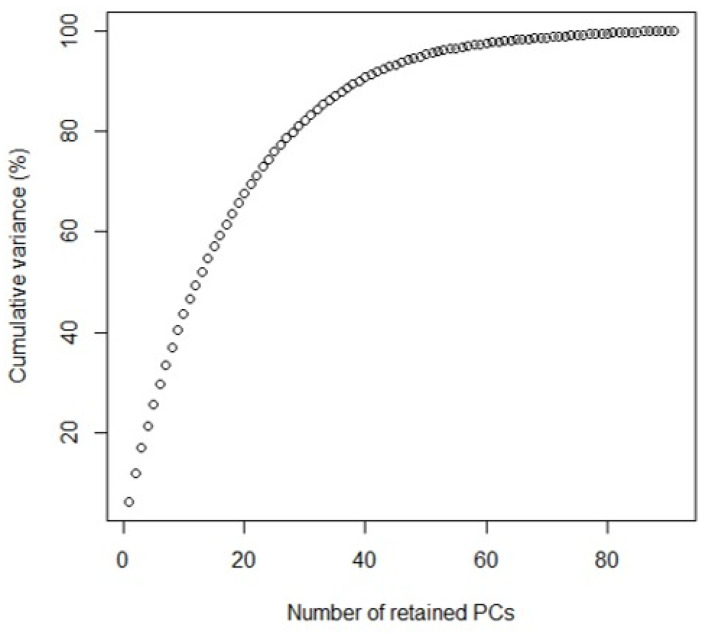
Graph of cumulated variance. Variance explained by PCA.

**Figure 10 plants-11-02803-f010:**
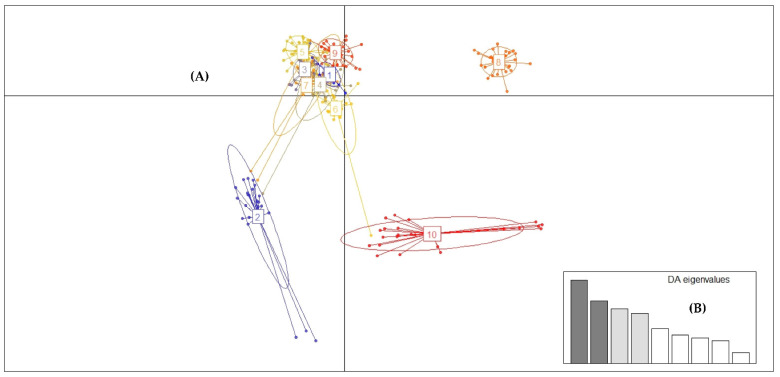
Analysis of the population genetic structure of *G. avellana* using discriminant analysis of principal components (DAPC) based on EST-SSR data. (**A**) Graph shows the formation of ten genetic groups. (**B**) Discriminant analysis eigenvalues. The bar plot of eigenvalues retains 40 PCs that explain over 80% of the cumulative variance.

**Figure 11 plants-11-02803-f011:**
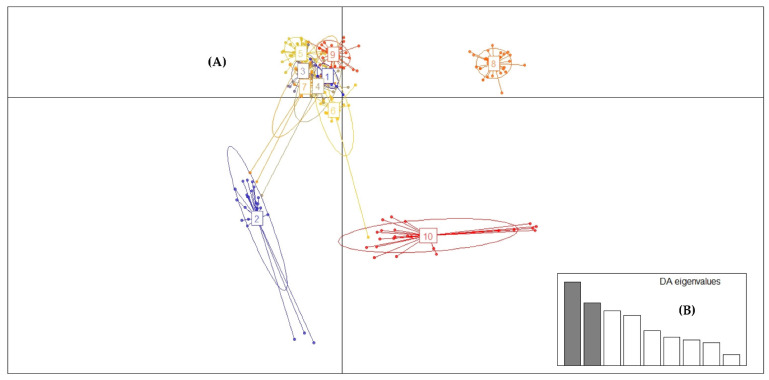
Analysis of the population genetic structure of *G. avellana* using discriminant analysis of principal components (DAPC) based on EST-SSR data. (**A**) Graph shows the formation of ten genetic groups. (**B**) Discriminant analysis eigenvalues. The bar plot of eigenvalues retains 80 PCs that explain 100% of the cumulative variance.

**Figure 12 plants-11-02803-f012:**
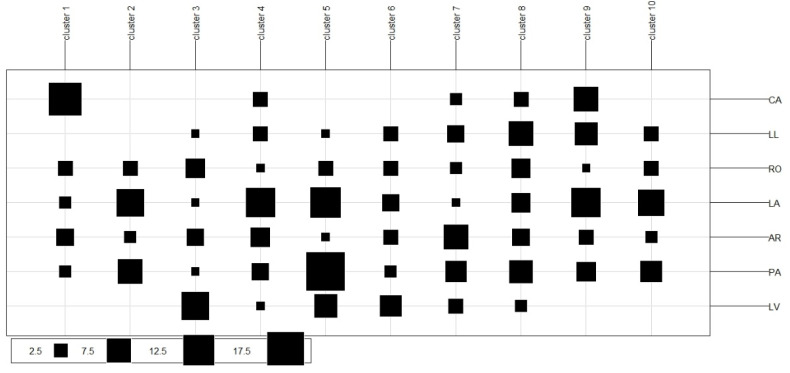
Genetic structure distribution graph. The number of individuals belonging to each sampled population is observed. La Veguilla (LV), Las Lomas (LL), Los Avellanos (LA), Punta de Águila (PA), Armerillo (AR), Roblería (RO), and Canelillo (CA).

**Figure 13 plants-11-02803-f013:**
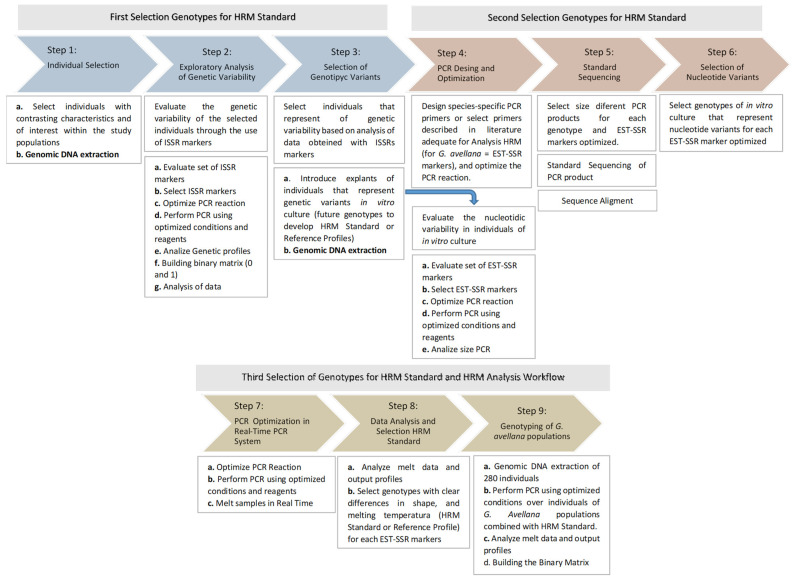
Scheme experimental workflow described for the genetic analysis of *G. avellana* Mol.

**Figure 14 plants-11-02803-f014:**
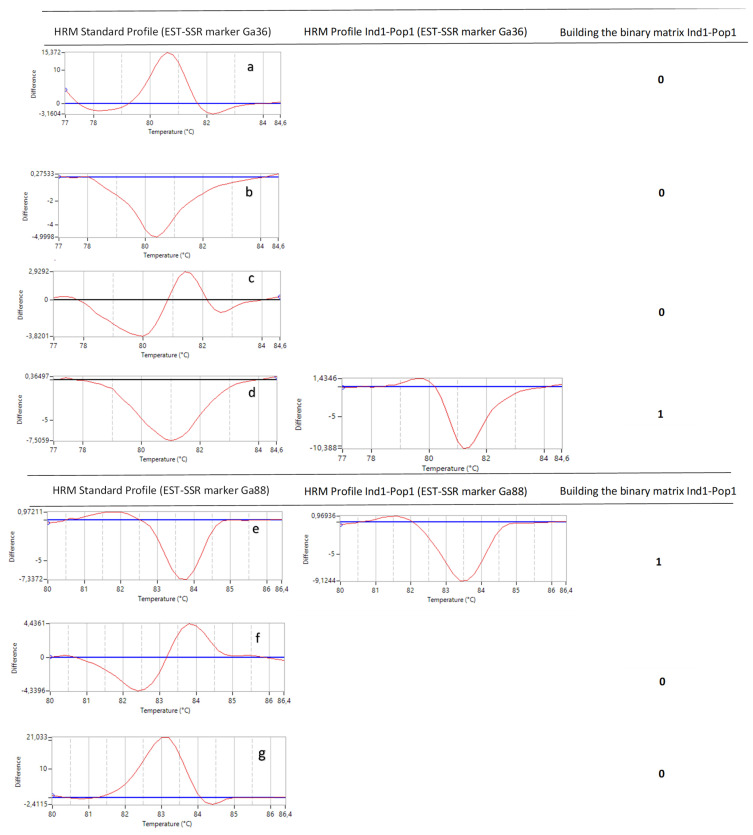
Scheme to building the binary matrix. The profiles on the left show HRM standard profile (reference profile) of genotypes of *G. avellana* Mol. analyzed using two EST-SSR markers (Ga36 and Ga88)**.** (**a**) Genotype *G. avellana* code 5R01, (**b**) Genotype *G. avellana* code 19RS01, (**c**) Genotype *G. avellana* code 8L01, (**d**) Genotype *G. avellana* code 43C01, (**e**) Genotype *G. avellana* code 38P01, (**f**) Genotype *G. avellana* code 27A02, and (**g**) Genotype *G. avellana* code 21C02. The center image shows the profile obtained for one individual (Ind1) of one population (Pop1). The image on the right shows the transformation to 0 and 1.

**Table 1 plants-11-02803-t001:** Plants material collected sites of the seven populations of *G. avellana* Mol. used in the study. Location distributed from north to south: La Veguilla (LV), Las Lomas (LL), Los Avellanos (LA), Punta de Águila (PA), Armerillo (AR), Roblería (RO), and Canelillo (CA).

Populations Code	Mountain	Coordinates		Elevation (masl)
		W	S	
LV	Andes	−35.3828573	−71.1113449	900–980
LL	Andes	−35.4730431	−71.1769904	850–890
LA	Andes	−35.5629722	−71.1587165	1200–1290
PA	Andes	−35.604411	−71.074437	1301–1420
AR	Andes	−35.7034586	−71.1104523	520–613
RO	Andes	−35.8602489	−71.271781	599–614
CA	Costa	−35.9813956	−72.649387	240–320

**Table 2 plants-11-02803-t002:** Permutation importance of variables for the Maxent Model.

Code	Environmental Variables	Units	Percent Contribution
Bio1	Annual Mean Temperature	°C	43.7
Water Steam	Water vapor pressure	kPa	23.4
Bio15	Precipitation Seasonality (Coefficient of Variation)	Percent	7
Bio17	Precipitation of Driest Quarter	mm	5.3
Bio13	Precipitation of Wettest Month	mm	5.2
Bio3	Isothermality (BIO2/BIO7) (×100)	Percent	4.6
Wind	Wind speed	m s^−1^	4.3
Bio16	Precipitation of Wettest Quarter	mm	2.7
Radiation	Solar radiation	Kj m^−2^ day^−1^	1.4
Bio6	Min Temperature of Coldest Month	°C	1.3
Bio4	Temperature Seasonality (standard deviation ×100)	°C or Percent	1
Bio2	Mean Diurnal Range (Mean of monthly (max temp − min temp))	°C	0.1

**Table 3 plants-11-02803-t003:** Genetic diversity estimates at eight EST-SSR markers for seven populations of *G. avellana* Mol. N: sample size, Pa: Private alleles, Ne: effective number of alleles, G_ST_: Genetic differentiation coefficient between populations, %P: percentage of polymorphic loci, and HRM Profile = Maximum HRM profiles found in each population (not exclusive to each population).

Population	N	Pa	Ne	G_ST_	%P	HRM Profile
La Veguilla	29	49	1.45	0.0368	45.79	4
Las Lomas	30	34	1.31	0.0344	31.78	4
Los Avellanos	66	53	1.49	0.0361	49.53	5
Punta de Águila	60	59	1.55	0.0361	55.14	5
Armerillo	36	41	1.38	0.0358	38.32	4
Roblería	29	46	1.42	0.0361	42.99	3
Canelillo	30	20	1.18	0.0339	18.69	3
Total	280		9.78	0.0431		28
Average		43	1.40	0.0356	40.32	4

**Table 4 plants-11-02803-t004:** Summary of molecular variance (AMOVA) from EST-SSR loci data collected of the matrix of genetic distances of 280 genotypes from 7 populations of *Gevuina avellana* Mol.

Source of Variation	df	SSD	CV	% Total	Fixation Index
Among populations	6	65.716	10.953	5	Fst = 0.052 **
Within population	273	945.030	3.462	95	
Total	279	1010.746		100	

df: degrees of freedom; SSD: sum of squares; CV: variance component estimates; % Total: percentage of total variance contributes by each component. ** *p* < 0.001 (significance test after 999 permutations).

## Data Availability

All data included in the main text.

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
