# Peer review of "Application of MaxEnt Modeling and HRM Analysis to Support the Conservation and Domestication of *Gevuina avellana* Mol. in Central Chile"

_plants, 2022, doi:10.3390/plants11202803_

Round 1

Reviewer 1 Report

In this manuscript authors applied two approaches viz. the MaxEnt model to predict the  current and future potential distribution coupled with HRM analysis to assess its genetic diversity  of Gevuina avellana and understand how the species would respond to these changes under climate change, Study is quite interesting however few lacunas make it difficult to understand it clearly.

Introduction described the maxent model and HRM analysis. However there is no mention of how these two are integrated? Objective is not clearly defined. Lot of introductory material is seen in results

Materials and Methods

4.1  section can be shifted to introduction

Line 465: Adaptation to extreme edaphoclimatic zones and agronomic rate  characteristics (for example age, fruit size, high fruit production year to year). Do you have the data for these if so it can be presented to know the variability.

4.3.3. No details of model about the calibration and validation. ENM eval or Kuenm can be used instead of the default model running which makes the results more reliable. For details refer to Cobos et al Peer J 2019 and Hebbar et al. Plants 2022..

Line number:483 For predicting the current distribution 30 arc sec climate data was used while for future 2.5 minute was used. Why selected different resolution data for different scenarios?

Line number: 486-488 The reason for selecting the future global climate models (GCMs): CNRM-CM6-1 and MIROC-ES2L not mentioned.  Is there any reference in support of the selections?

Line 511 to 517 include in introduction

Line 519 to 520: contrasting phenological and  phenotypic characteristics of different kinds (differences in flowering time, tolerance to  stress, fruit size, among others). This information of 7 population is very useful for writing discussion.

Line 521: mentions explants of tissue culture. It is not clear. Elaborate it.

Line 523: these genotype- Are they genotypes?

Results and discussions:

2.1 can be included in introduction

2.3. MaxEnt Model of Distribution potential current and future of G. avellana instead of that current and future distribution potential of G. avellana

Line number-125-134 and again Line number-136-138 should be included in introduction.

Fig. 4 title to be revised

Line 176 to 187: to be included in introduction

Line 188 to 201: scope for language improvement and need more clarity    

Comment 2: Line number- 140-142 –Is there any criteria has taken for classification of suitability. Also there is discontinuity found in the classification

Habitat suitability class for current (Fig 3) and future projections are different and non-continuous. Why this difference and it would be difficult to predict the shift.

Table 2 Please recheck the units. Bio3, bio4 and bio 15

The legend  of fig 5 is not visible

208 to 212: include in introduction

214 to 226 shift to introduction

Line 233: genotypes check?

Figure 7. Discrimination of G. avellana genotypes- check if it is genotypes

Line 255 genotypes

Discussion

Discussion is too general and it is not interpreted based on the results of the study. How climate change is affecting the distribution of the crop and based on the HRM analysis what population is suitable for a particular climate should have been suggested.

Conclusion is also too general and not specific the the outcome of the study and needs to be changed.

General comment: scope for improving English.

Author Response

Response to Reviewer 1 Comments

Point 1: In this manuscript authors applied two approaches viz. the MaxEnt model to predict the  current and future potential distribution coupled with HRM analysis to assess its genetic diversity  of Gevuina avellana and understand how the species would respond to these changes under climate change, Study is quite interesting however few lacunas make it difficult to understand it clearly.

Introduction described the maxent model and HRM analysis. However there is no mention of how these two are integrated? Objective is not clearly defined. Lot of introductory material is seen in results 

Response 1:

 I do appreciate your words in considering the research. Based on your comments, I included most of the suggested changes, so I hope I have corrected the gaps that made it difficult to understand. I re-ordered the misclassified sections, deepened the discussion, and incorporated new components and figures in methods and results to explain the methodology and strategy used in the research.

Point 2: Adaptation to extreme edaphoclimatic zones and agronomic rate  characteristics (for example age, fruit size, high fruit production year to year). Do you have the data for these if so it can be presented to know the variability.

Response 2:

In this article, I will not include the phenotypic, phenological, and edaphoclimatic criteria   because they are part of another  paper in development, where I consider  all these antecedents in-depth.

Point 3: No details of model about the calibration and validation. ENM eval or Kuenm can be used instead of the default model running which makes the results more reliable. For details refer to Cobos et al Peer J 2019 and Hebbar et al. Plants 2022. 

Response 3: 

At this point, I incorporated details into the discussion. As for the suggested article, it is very absorbing. Unfortunately, my current level of knowledge in the modeling area prevents me at this time from incorporating and delving into the suggested methodology.

Point 4: For predicting the current distribution 30 arc sec climate data was used while for future 2.5 minute was used. Why selected different resolution data for different scenarios?

 Perhaps at this point, there is a writing problem on my part. The current and future models consider the following :

1.- For current distribution, the data used was 30 arc. because of its greater resolving power.

2.- For the development of future models, we require a comparison with the current distribution at the same resolution level. Therefore, the current distribution is made in 2.5 min, and the future distribution with each model, as well.

Reviewer 2 Report

Review of "Application of MaxEnt Modeling and HRM Analysis to Support the Conservation and Domestication of Gevuina avellana Mol. in Central Chile" for MDPI Plants.

This study analyzes the genetic structure of Gevuina avellana, a Southamerican wild relative of the well known Macadamia trees which has also commercial potential, and the predicted shifts of its potential area of distribution under several scenarios of climatic and anthropogenic change. All this with the perspective of contributing to both the conservation and the domestication of the species.

The study is based on Maxent models of species distributions and High-Resolution Melting analysis (HRM) of genetic variation of seven of the northernmost populations of the species.  HRM is a fast, low-cost technique that may successfully replace more expensive molecular analyses and which could benefit conservation genetic studies. Therefore I find this study quite interesting and I think that it may appeal to a wide audience of MDPI-Plants readers with interests in plant conservation.

Major comments:

In my opinion, the paper needs some refurbishing before it can be published. I provide here, as well as in an annotated pdf, specific details of what needs to be addressed.

First, the content should be fit to the structure of the usual papers in this journal (i.e., Introduction, Results, Discussion, Methods). In the current version of the manuscript there are many paragraphs that are in the wrong section or subsection.

Second, some of the Methods and the Results need more detail in order to clarify aspects that remain obscure. In the case of HRM, a diagram explaining the steps from sampling the 278 trees to building the "binary matrix" employed for genetic analyses would shed light on e.g. what do you consider an allele. A list indicating explicitly how many HRM curves you found for *each* of the eight markers employed (you say in L232  that you found 28 HRM profiles in total, but not which one were associated to each marker).

Finally, the Discussion should be completely revised. In the current version there is a lot of very interesting information, but most of it not explicitly related to the results of the paper, which are also scarcely discussed. Discussing the results obtained in your analyses and relating them to all the information that you provide would considerably enhance the manuscript.

Minor coments

Abstract

L18: "coupled with  High-Resolution Melting Analysis (HRM)"

L25: "a percentage of polymorphic loci"

Introduction

L 49: There is some production of macadamia honey. What about gevuina honey?

L60-64: This paragraph describes the justification for the paper. It should explain also the rationale under the methodological approach (genetic inclusive) chosen. I propose rewriting it, for example like this:

"The United Nations Assembly established 17 Sustainable Development Goals (SDGs) for the year 2030 and has included goals related to restoration and reforestation on its agenda [21]. A successful reforestation program should consider the environmental and genetic factors that influence plant performance (INCLUDE APPROPRIATE REFERENCE HERE). Climate is one of the critical factors influencing vegetation's type and distribution globally [19]. It is therefore essential understanding the far- reaching effects of climate change and the associated anthropogenic impacts [20]."

L67-68: remove reference 23 and place reference 22 where ref. 23 was (ref. 23 is unnecessary here).

L69: Reference 24 is not about the effect of climate change on plants and insects. I think that you mean ref 19 here. By the way, you may remove "and pest insects" as your study is only about the effects of CC on the distribution of plants.

L 72: Maybe you must explain that SSR are simple sequence repeats

L81: Here, at the end of the introduction make explicit what are your specific objectives and/or hypotheses in this study.

Results

L89-92: Delete. This is more appropriate fro th "Conclusion" that for the Results.

L106-107: Explain (in the Discussion section) whether you have also georeferenced points for trees in the National Reserves and whether the fact of not including them in the Maxent models could affect the results about the variables affecting G. avellana distribution.

L118: You write repeatedly throughout the manuscript "La veguilla". I wonder if it would be more correct "La Veguilla".

L120 (around): Tabla 1: Delete "UTM" from the table: these are lon-lat coordinates, not UTM.

L121: "Maxent model of current and future distribution of G. avellana""

L128-139: Remove this from here (this is the Result subsection). You may include it in the "Methods" section.

L146-147 (Figure 3): As the colour scale does not coincide with the values selected to define the different suitability areas, remove the reference to them in the legend of the figure. You may alternatively explain that the closest to "1", the highest is the suitability for G. avellana.

In addition, see comments fro methods (L470-477): You should include in the left map the Argentinean provinces were G. avellana is known to occur.

L149-152: Remove this. You already said the same in L139-140.

L 163-164: Move this to "Methods".

L168 Figure 4 A is not very much informative and can be included as a supplement.

L173-187: Remove this from here (this is the Result subsection). You may include it in the "Methods" section.

L208-212: Remove this from here (this is the Result subsection). You may include it in the "Methods" section.

L213-228: Remove this from here (this is the Result subsection). You may include it in the "Methods" section.

L232: Clarify whether these "28 HRM profiles" are also "HRM Standards".

L232: A table with the number of "alleles" or profiles for each of the eight markers (or even, for the number of alleles for each marker in each population) would enhance the paper. Place it around here.

L237-239: Explain what means the label  "RFU" in the y-axis. Explain also why some genotypes are displayed in blue and others in red.

L242 (around) Figure: 7: Could it be possible either displaying a legend with the colours and the genotype/populations or shifting some of the colours chosen? I'm unable to distinguish between "red" and "burgandy" nor between "sky-blue" and "light-blue".

L243-247(legend of Figure 7): I think that these should be called "alleles" instead of "genotypes". Please, confirm here or in the text that this particular marker GA38 has only one (and a different one) allele in each of the 7 studied populations (In fact, Diaz 2010 , in her Tabla 4 says that this marker was monomorphic in her study)..

L251 Table 3: This table puzzles me a bit. What is tabulated in the column "HRM profile"? Are they HRM profiles *exclusive* of each population? Clarify.

By the way, if you only detected 28 HRM profiles for the whole eight markers, how could there be 303 private alleles?

In addition: Replace the commas for decimal points in the figures for  Total Ne and for Average Gst

L257-261: These are not your results. Place this comment in the Discussion.

L266-268: See comment above (L251)  about private alleles in Table 3.

L278-281: These are not your results. Place this comment in the Discussion.

L282-285: I think that this should be placed in subsection 4.4.4.Data Analysis, in Methods.

L288-289: It would be very useful that you provide a plot similar to Figure 6 of Jombart et al. 2010 showing the variation of BIC with the number of clusters when performing the k-means cluster selection. In addition, you should describe the amount of variation accounted by the two principal components that you use to plot the clusters in Fig. 8 (it seems that the third and fourth eigenvalues are almost as large as the second one). You should also comment how many eigenvalues were considered (retained)  in the DAPC (the inset in fig 8 Suggests that you included 8, but you should write down this explicitly in the text).

L288-289: It is kind of a shocking result that you detected 10 genetic populations over 7 geographic populations. You should revise your Methods and if everything is ok you shoud  discuss (in the Discussion) what are the possible cuses for this. Maybe the sampled populations are not entirely natural and have been artificially mixed by the peasants introducing in each locality genotypes from some ancestral original populations with some desired phenotypic characteristics?

L290-293: These are not your results. Place this comment in the Discussion.

Discussion

L309-437: There is lot of interesting information here, but there is almost not discussion of the results obtained in your study. Discussing the results obtained in your analyses and relating them to all the information that you provide would considerably enhance the manuscript. If you includes some specific objectives in the Introduction (see suggestion for L81) this would be very easy.

Material and Methods

L441-443: delete and substitute for " and temperatures range from 30º C in summer to -8ºC in winter"

L464-466. I think that this is redundant with L528-521. Describe here in detail the criteria employed to select the 287 trees (see comment for L528-521 below).

L466-468: Delete this. This information appears in other places in the manuscript.

470-477: In order for the Maxent model to be more robust, you should include also the known distribution of Gevuina in Argentina. Just include the Argentinean localities available in GBIF (or, if there is none, explain it here). Logically, include also in your study (an in the maps that you show in the Figures) the Argentinean provinces and project the future distribution in these provinces as well.

L490-492: Reiterative. Delete.

L497-499: The citation for Maxent should be  "Phillips et al. (2022)", and in the reference list it should appear as:

Steven J. Phillips, Miroslav Dudík, Robert E. Schapire. [Internet] Maxent software for modeling species niches and distributions (Version 3.4.4). Available from url: http://biodiversityinformatics.amnh.org/open_source/maxent/. Accessed on 2022-8-24.

L504 (subsection " Genotyping with High-Resolution Melting Analisys ") Please, include a diagram showing all the steps from extracting DNA to "binary matrix" (l554) construction.

L516-517: This is not a "phylogenetic study" but a "conservation genetics study"

L525: Add numeric citation for "Diaz (2010)"

L518-521: The phenotypic criteria employed to select individuals within populations is critical to understand the genetic results. Please, explain (e.g., tabulate)  the range of phenotypic values within each population for al traits mentioned and for those not mentioned in this paragraph (i.e., the "among others").

L527: Maybe you can tabulate the sizes of the PCR fragments.

L536: G. avellana (in italics).

L550: "Two hundred and eighty trees" (or individuals)

L552: Several "Binary matrices" or just one binary matrix (L554)? Please, clarify. Indicate also the dimensions of the matrix, i.e., 278 rows x how many columns?

L557: Name the functions in the adegenet package employed in the analysis, and the parameters (arguments) used with each one (if they were different than the default arguments). At least you should cite find.clusters() and dapc() but you probably used some other functions.

References

L740: The URL for this reference is wrong (is the URL of another paper). The correct one is  https://doi.org/10.1186/1471-2156-11-94

Author Response

Response to Reviewer 1 Comments

Point 1: The study is based on Maxent models of species distributions and High-Resolution Melting analysis (HRM) of genetic variation of seven of the northernmost populations of the species.  HRM is a fast, low-cost technique that may successfully replace more expensive molecular analyses and which could benefit conservation genetic studies. Therefore I find this study quite interesting and I think that it may appeal to a wide audience of MDPI-Plants readers with interests in plant conservation.

 Response 1:

Thank you very much for your comments and suggestions, it has been an honor and a great help to receive such detailed comments that allowed us to substantially improve the article.

Point 2: First, the content should be fit to the structure of the usual papers in this journal (i.e., Introduction, Results, Discussion, Methods). In the current version of the manuscript there are many paragraphs that are in the wrong section or subsection.

Response 2:

You are right. It was a mistake not to fully respect the format established by MDPI. In this regard, I incorporated most of the suggested changes. Therefore, I hope to have corrected most of these editing errors.

Point 3: Second, some of the Methods and the Results need more detail in order to clarify aspects that remain obscure. In the case of HRM, a diagram explaining the steps from sampling the 278 trees to building the "binary matrix" employed for genetic analyses would shed light on e.g. what do you consider an allele. A list indicating explicitly how many HRM curves you found for *each* of the eight markers employed (you say in L232  that you found 28 HRM profiles in total, but not which one were associated to each marker).

Response 3:

Based on your comments,  I included most of the suggested changes, so I hope I have corrected the gaps that made it difficult to understand.  I re-ordered the misclassified sections, deepened the discussion,  and incorporated new components and figures in methods and results to explain the methodology and strategy used in the research.

Point 4: Finally, the Discussion should be completely revised. In the current version there is a lot of very interesting information, but most of it not explicitly related to the results of the paper, which are also scarcely discussed. Discussing the results obtained in your analyses and relating them to all the information that you provide would considerably enhance the manuscript.

Response 4:

New discussion was incorporated according to the requested criteria.

Point 5. In addition, see comments fro methods (L470-477): You should include in the left map the Argentinean provinces were G. avellana is known to occur.

Response 5:

This point was not developed because it is not part of the objectives of this study. In addition, there is no reliable and sufficient registry data to develop distribution maps of G. avellana in Argentina. GBIF is a database that incorporates records from different sources of information, museums, naturalists, and amateurs. Therefore, the information can be underestimated. For example, G. avellana in the GBIF database indicates the presence in the United States, Argentina, Peru, and New Zealand, but the origin of the records is mostly unknown.

Point 6. The phenotypic criteria employed to select individuals within populations is critical to understand the genetic results. Please, explain (e.g., tabulate)  the range of phenotypic values within each population for al traits mentioned and for those not mentioned in this paragraph (i.e., the "among others").

 Response 6:

 In this article, I will not include the phenotypic, phenological, and edaphoclimatic criteria  because they are part of another  paper in development, where I consider  all these antecedents in-depth.

Round 2

Reviewer 1 Report

Most of the queries were addressed and  the flow is much better. Still there is scope for further improvement. Discussion is too lengthy. There is no need to define and highlight the advantage of each technique. Just if if it is refered it is ok. Some of the corrections are highlighted here.

Line 82: and gamut), is it gama?

Line 135: Future of a specie [49]. It is species

Line No 201 to 235: 2.1. Study Area and Plant Materials   should go to material and methods and it is not part of the results.

Line 244: MaxEnt Model of Distribution instead  MaxEnt Model projection of

Line non: 264 Validation of our model allowed us to arrive at AUC values of 0.953

Line from 264 to 269 recheck. It is not complete

Line 306: Maxent model current for - is it current projection?

Line 642 and 643 not clear. Rewrite.

Subtitles 2.2. and 4.2 are same. They can be modified

Line 1063: First, the complete model with all the variables and the default  configuration the most influential variables. Not complete?

.43.2. Selection Molecular Markers. Selection of

Author Response

I am afraid I only disagree with this comment, so I would appreciate it if you allow me to maintain this section where it is now.

Point 1:  Line No 201 to 235: 2.1. Study Area and Plant Materials   should go to material and methods and it is not part of the results.

Response 1:

I kindly request that you allow me to keep this section in its place since the description there does correspond to the results.

As mentioned on several occasions, in Central Chile, there is almost complete ignorance of G. avellana. Therefore, the description and number of populations detailed in the section, the preparation of the maps, and the interaction with the communities mentioned in the opening paragraph, are the results of the study carried out in a particular geographical area (Maule Region).

Reviewer 2 Report

Review of "Application of MaxEnt Modeling and HRM Analysis to Support the Conservation and Domestication of Gevuina avellana Mol. in Central Chile" for MDPI Plants.

The authors have considered most of the issues and suggestions raised in my previous review. The paper has improved a lot. Only a few amendments remain to be made.

In the following, I use the line numbers from the corrected pdf.

Minor coments

Abstract

L25: "showing a percentage of polymorphic loci"

Results

L244: This subsetion should be entitled  "Maxent model of current and future distribution of G. avellana""

L493 Table 3: Explain in the legend of the Table whether  "HRM profile" are HRM profiles *exclusive* of each population or are another thing.

Discussion

L640: "The current and future potential distribution of G. avellana described in the present"

L640-665: place this in "Methods"

L696-697: "twelve variables that are not correlated (Figure 4C,"

L731-753: Remove this paragraph. It is unnecessary for the Discussion.

L759-L774: (From "In terms of describing..." to "...-Step 9.". Place this paragraph (if necessary) in Methods.

Material and Methods

L1148: Name the functions in the adegenet package employed in the analysis, and the parameters (arguments) used with each one (if they were different than the default arguments). At least you should cite find.clusters() and dapc() but you probably used some other functions.

Author Response

I'm afraid I have to disagree specifically with only two of the comments. For this reason, I request that you allow me, as far as possible, to keep these two sections in the place where they are currently.

Point 1:   L640-665: place this in "Methods"

Response 1:

This introduction will facilitate an understanding of how to address the flow of information and the parameters that should be considered when using this methodology. For this reason, I suppose it should be kept in the discussion.

Point 2:   L731-753: Remove this paragraph. It is unnecessary for the Discussion.

Response 2:

In the same way, I request that you allow me to keep this paragraph since, according to my experience, it contributes significantly to the discussion. Moreover, it provides a relevant background that researchers must consider for developing ecological niche studies.
